# Transcriptomically Revealed Oligo-Fucoidan Enhances the Immune System and Protects Hepatocytes via the ASGPR/STAT3/HNF4A Axis

**DOI:** 10.3390/biom10060898

**Published:** 2020-06-12

**Authors:** Chun-Chia Cheng, Wan-Yu Yang, Ming-Chen Hsiao, Kuan-Hao Lin, Hao-Wei Lee, Chiou-Hwa Yuh

**Affiliations:** 1Institute of Molecular and Genomic Medicine, National Health Research Institutes, Zhunan 35053, Taiwan; cccheng.biocompare@gmail.com (C.-C.C.); cs081011@nhri.edu.tw (W.-Y.Y.); khlin@nhri.edu.tw (K.-H.L.); haowei0115@nhri.edu.tw (H.-W.L.); 2Radiation Biology Research Center, Institute for Radiological Research, Chang Gung University/Chang Gung Memorial Hospital at Linkou, Taoyuan 33302, Taiwan; 3Research and Development Center, Hi-Q Marine Biotech International Ltd. Songshan District, Taipei 10561, Taiwan; ming.hsiao@hiqbio.com; 4Institute of Bioinformatics and Structural Biology, National Tsing-Hua University, Hsinchu 30013, Taiwan; 5Department of Biological Science and Technology, National Chiao Tung University, Hsinchu 30010, Taiwan; 6Program in Environmental and Occupational Medicine, Kaohsiung Medical University, Kaohsiung 80708, Taiwan

**Keywords:** oligo-fucoidan, hepatocyte nuclear factor 4 alpha (HNF4A), hepatocyte

## Abstract

Oligo-fucoidan, a sulfated polysaccharide extracted from brown seaweed, exhibits anti-inflammatory and anti-tumor effects. However, the knowledge concerning the detailed mechanism of oligo-fucoidan on liver cells is obscure. In this study, we investigate the effect of oligo-fucoidan in normal hepatocytes by transcriptomic analysis. Using an oligo-fucoidan oral gavage in wild-type adult zebrafish, we find that oligo-fucoidan pretreatment enhances the immune system and anti-viral genes in hepatocytes. Oligo-fucoidan pretreatment also decreases the expression of lipogenic enzymes and liver fibrosis genes. Using pathway analysis, we identify hepatocyte nuclear factor 4 alpha (HNF4A) to be the potential driver gene. We further investigate whether hepatocyte nuclear factor 4 alpha (HNF4A) could be induced by oligo-fucoidan and the underlying mechanism. Therefore, a normal hepatocyte clone 9 cell as an in vitro model was used. We demonstrate that oligo-fucoidan increases cell viability, Cyp3a4 activity, and Hnf4a expression in clone 9 cells. We further demonstrate that oligo-fucoidan might bind to asialoglycoprotein receptors (ASGPR) in normal hepatocytes through both in vitro and in vivo competition assays. This binding, consequently activating the signal transducer and activator of transcription 3 (STAT3), increases the expression of the P1 isoform of HNF4A. According to our data, we suggest that oligo-fucoidan not only enhances the gene expression associated with anti-viral ability and immunity, but also increases P1-HNF4A levels through ASGPR/STAT3 axis, resulting in protecting hepatocytes.

## 1. Introduction

The liver is the biggest organ in the body, and there are many factors that can cause liver damage and lead to liver diseases [1], which is a progressive process from inflammation, steatohepatitis, fibrosis, and cirrhosis, to eventually liver cancer. Effective and therapeutic means of preventing liver diseases and cancer formation are urgently needed [2]. Oligo-fucoidan (OF), a sulfated polysaccharide extracted from brown seaweed, exhibits anti-cancer and anti-proliferation effects in various cancer types, including liver cancer [3]. Multiple signal transduction pathways are related to its anti-cancer effects, such as mitogen-activated protein kinase (MAPK), extracellular signal-regulated kinase (ERK), phosphatidylinositol 3′-kinase (PI3K), protein kinase B (PKB, AKT), mechanistic target of rapamycin (mTOR), transforming growth factor beta 1 (TGF-beta 1), and small mothers against decapentaplegic (SMAD) [3,4]. In addition, fucoidan enhances the ubiquitin-dependent degradation of transforming growth factor-beta receptors (TGFRs) in lung and breast cancer, also inducing reactive oxygen species (ROS) and C/EBP homologous protein (CHOP) through Toll-like receptor 4 (TLR4) in lung cancer, additionally inhibiting the vascular endothelial growth factor (VEGF) receptor 2 (VEGFR2)/ERK/VEGF signaling pathway in anti-angiogenesis [3].

Besides, fucoidan has been found to prevent CCl4-induced liver fibrosis [5] and attenuate N-nitrosodiethylamine-induced liver fibrosis [6]. However, the mechanism for fucoidan to protect hepatocytes is not clear. A previous study demonstrated that OF is a potential agonist for C-type lectin-like receptor 2 in platelets [7]. Since OF is a mixture containing galactose, we proposed that it could bind to the major surface lectin receptors such as asialoglycoprotein receptors (ASGPR) in hepatocytes [8]. In addition, the galactose exposed by desialylated platelets are recognized by ASGPR in hepatocytes and activates the phosphorylation of STAT3 (pSTAT3) [9]. Therefore, we assumed that OF can bind to ASGPR to activate STAT3 in normal hepatocytes for cyto-protecting effect, since the activation of STAT3 prevents acute liver injury, fatty liver disease, and alcoholic hepatitis [10].

Research on disease mechanisms and the development of new drugs cannot solely rely on in vitro systems, whereas in vivo animal models are also required. Therefore, we investigate and uncover the differential genes in a zebrafish model, because the cost to maintain animals such as mice is significantly higher. Zebrafish is an emerging animal model and has many advantages, such as external fertilization, external development, a short sexual maturation time, a lower cost, and the fact that its genome exhibits 87% homology to the human genome [11,12]. Therefore, zebrafish has become a popular model for studying human diseases [13]. The use of zebrafish as a human disease model has become increasingly common [14]. Zebrafish hepatocellular carcinoma (HCC) models deepen our understanding of cancer formation via genetic, environmental, and metabolic insults [15,16,17,18,19,20,21,22]. Here, we investigate the beneficial effect of OF in hepatocytes and report a new possibility of OF for strengthening the immune system and hepatocellular functions in wild-type zebrafishes. Moreover, we uncover the liver protective effect of OF and reveal the mechanism of binding to the ASGPR, activating STAT3, and upregulating the P1 isoform of hepatocyte nuclear factor 4 alpha (HNF4A) in normal hepatocyte clone 9 cells. 

## 2. Materials and Methods

### 2.1. Oligo-Fucoidan (OF)

The low molecular weight oligo-fucoidan was provided by Hi-Q Marine Biotech International Ltd. (Taipei city, Taiwan). It was extracted from brown seaweed with enzyme hydrolysis to an average molecular weight in a range between 500 and 1500 Da.

### 2.2. Zebrafish Maintenance

The AB zebrafish strain (*Danio rerio*) was obtained from the Zebrafish International Resource Center (ZIRC, Eugene, OR, USA) and used in this study. All experiments were approved by the Institution Animal Care and Use Committee (IACUC) of the National Health Research Institutes (NHRI, protocol number NHRI-IACUC-106118-A).

### 2.3. Oligo-Fucoidan Administration in Adult Wild-Type Zebrafish and Tissue Collection

Oral gavaging was conducted as described previously [23,24]. The fish were anesthetized using MS-222 solution (150 mg/L), and 5 μL of solution was injected slowly into the intestinal tract of adult fish using flexible tubing. Here, five-month-old wild-type fish (*N* = 40) were divided into two groups, either no treatment (mock) or OF administrations via an oral gavage with OF (300 mg/kg) thrice a week for one month. After one month of oral feeding, the fish were sacrificed, and the hepatic tissue was immediately frozen in liquid nitrogen and stored at −80 °C for later RNA isolation and quantitative polymerase chain reaction (qPCR) analysis. Two batches of wild-type fish fed with OF were performed.

### 2.4. Total RNA Isolation and cDNA Synthesis

Total RNA isolation and complementary DNA (cDNA) synthesis was conducted as described previously [25]. The RNA was isolated using a NucleoSpin^®^ RNA kit (Macherey-Nagel, Duren, Germany). The RNA samples were stored at −80 °C. The cDNA was synthesized using an iScriptTM cDNA Synthesis Kit (Bio-Rad, Hercules, CA, USA). The cDNA was stored at −20 °C for long-term preservation.

### 2.5. GeneTitan^™^ Array for Gene Expression Profiling

ZebGene 1.1 ST Array Plates (Affymetrix, Santa Clara, CA, USA) were used for the whole-genome transcriptome analysis. Transcriptome analysis console (TAC) software (version 4.0.2) (Affymetrix) was used for the data processing including normalization, probe summarization, and data quality control. Guanine cytosine count normalization (GCCN) and signal space transformation (SST) algorithms were used for normalization to adjust CytoScan raw data (CEL files) intensities, allowing inter-platform comparisons. After we used TAC for data interpretation, differentially expressed genes of OF were identified for both the treated and control samples. The expression analysis settings were as follows: gene-level fold change <−2 or >2, gene-level *p*-value <0.05, analysis of variance (ANOVA) method: Ebayes. The raw data of the microarray have been submitted to the National Center for Biotechnology Information (NCBI) gene expression omnibus (GEO) (http://www.ncbi.nlm.nih.gov/geo/) under accession code GSE148324. 

The gene ontology analysis was performed using WebGestalt (WEB-based GEne SeT AnaLysis Toolkit, http://www.webgestalt.org/) and activated pathways were selected and matched according to the Kyoto Encyclopedia of Genes and Genomes (KEGG) database.

### 2.6. Quantitative Polymerase Chain Reaction (qPCR)

The qPCR experiments were performed as described previously [25] by using the KAPA SYBR^®^ FAST qPCR Master Mix (Roche, Basel, Switzerland) and ABI Real-Time PCR System. The information for primers used in the qPCR are listed in Table 1.

The analysis of the qPCR results were performed as described previously [25], by normalization to actin as an internal control, and calculated the fold change between wild-type zebrafish with or without oligo-fucoidan feeding. All the experiments were performed three times, each time with triplicated samples. The median and standard deviation were calculated from three sets of qPCR experiments.

### 2.7. Cell Viability and Gene Knockdown

Cell culture condition and cell viability assay were performed as described previously [26]. Cell viability was determined by using WST-1 assay (Water Soluble Tetrazolium Salts) (Takara Bio., Kusatsu, Shiga, Japan) and performed at least three independent batches. The knockdown of *HNF4A* was conducted using a short-hairpin RNA (shRNA) (target sequence: CGAACAGATCCAGTTCATCAA) in the vector pLKO.1-puro using a lentivirus system. The knockdown of ASGPR1 was conducted using two different shRNA (target sequence: GCACCACATAGGCCCTG-TGAA and ACGTGAAGCAGTTCGTGTCTG) in the vector pLKO.1-puro using a lentivirus system. All shRNA were purchased from the National RNAi Core Facility of Academia Sinica (Taipei, Taiwan). 

### 2.8. Western Blots

Western blots analyses were performed as described previously [25] using OF-treated Clone 9 cells, while immunoreactive proteins were detected by an enhanced chemiluminescence kit (Bio-Rad). The specific antibodies against STAT3 (signal transducer and activator of transcription 3), phosphorylated STAT3 (pSTAT3), HNF4A (hepatocyte nuclear factor 4, alpha), and GAPDH (glyceraldehyde 3-phosphate dehydrogenase) were purchased from Cell Signaling Technology, Inc. (Danvers, MA, USA).

### 2.9. Flow Cytometry for In Vitro Completion Assay

Clone 9 cells were trypsinized from the culture plates. Detached cells were rinsed with phosphate-buffered saline (PBS) and underwent 300 g centrifugation for 5 min totally thrice. Then, the cells were immersed into a staining buffer for 1 h at room temperature. The staining buffer was composed of fluorescein isothiocyanate-conjugated asialofetuin (ASF-FITC) with 1 mg/mL of OF or PBS, respectively. Subsequently, discarding the staining buffer, the cells were washed with PBS and centrifuged at 300 g for 5 min, totally thrice. Then, cells were resuspended into 1 mL of PBS and were analyzed by a FACSCalibur Flow Cytometer (Becton, Dickinson and Company (BD), Franklin Lakes, NJ, USA).

### 2.10. Asialoglycoprotein Receptor (ASGPR) Nuclear Imaging for In Vivo Competition Assay

The in vivo competition assay was performed as described previously [26]. Adult Bagg albino strain c (BALB/c) mice were purchased from the National Laboratory Animal Center and kept in clean conventional conditions. The animals were fed ad libitum. The ^68^Ga-NOTA-HL (^68^Ga-2,2’,2’’-(1,4,7-triazonane-1,4,7-triyl) triacetic acid-hexavalent lactoside) was synthesized by the Institute of Nuclear Energy Research (Atomic Energy Council, Yonghe Dist., New Taipei City, Taiwan) and was capable of binding to ASGPR in the body [27], which could be considered as the ligand for ASGPR. To confirm whether OF binds to ASGPR on hepatocytes in vivo, 100 mg OF and PBS was administrated respectively 1 h prior to the ^68^Ga-NOTA-HL intravenous injection. The injection dose of ^68^Ga-NOTA-HL was 200 ± 20 μCi. The in vivo competition studies were approved by the ethical review committee of the Institute of Nuclear Energy Research.

### 2.11. Measurement of CYP3A4 activity

The OF-treated and BBI608-treated clone 9 cells were collected respectively, and a commercial kit was used to measure CYP3A4 activity (BioVision, Inc., Milpitas, CA, USA). In brief, the assay utilizes a non-fluorescent CYP3A4 substrate that is converted into a fluorescent metabolite detected in the visible range (Ex/Em = 535/587 nm). The CYP3A4 activity was presented by percentage whereas the cells without treatment was set as 100%.

### 2.12. Statistical Analysis

The statistical analyses of the current study were all performed by GraphPad Prism v8.02 (GraphPad Software, Inc., San Diego, CA, USA). Here, *p*-values <0.05 were considered to be statistically significant and are presented as follows herein: *0.01 < *p* ≤ 0.05; **0.001 < *p* ≤ 0.01; ***0.0001 < *p* ≤ 0.001; *****p* ≤ 0.0001.

## 3. Results

### 3.1. Oligo-Fucoidan Increases Immune System and Downregulates Lipogenic and Fibrosis-Associated Genes

To observe the effect of OF on wild-type zebrafish, we administered OF (300 mg/kg) orally to five-month-old wild-type fish thrice a week in a one–month period. Then, liver tissue was excised for RNA extraction and cDNA synthesis. In order to explore OF-mediated gene expression in hepatocytes, the whole-genome expression profiles were analyzed by a GeneTitan array. Comparing two batches of OF-fed wild-type fish, we identified 124 genes that were upregulated at least two-fold with a *p*-value less than 0.05 (Figure 1A). The gene ontology analysis revealed that those upregulated genes were involved in immune system development (Figure 1B). Especially, *notch1a*, interleukin 7 receptor (*il7r*), interleukin-1 receptor-associated kinase 3 (*irak3*), cytochrome P450, family 2, subfamily R, polypeptide 1 (*cyp2r1*), TLE family member 3, transcriptional corepressor b (*tle3b*), and PTTG1 interacting protein B (*pttg1ipb*) were upregulated by OF treatment in the wild-type fish. Interestingly, we found the anti-viral gatekeeper *mxf* [28] and hepcidin, the host defense multiple microbial, *hamp,* in zebrafish [29], were also upregulated. The selected genes upregulated by OF from the microarray analysis are shown in Figure 1C. Currently, the world is experiencing the COVID-19 pandemic [30], and it will be very intriguing to see the upregulation of anti-viral, anti-microbial, and immune modulation proteins in OF-fed wild-type fish. Hence, we would like to verify these results using qPCR.

Comparing two batches of OF-fed wild-type fish, we identified 114 genes that were downregulated at least two-fold with a *p*-value less than 0.05 (Figure 2A). The pathway analysis revealed that genes involved in fatty acid metabolism, the peroxisome proliferator-activated receptors (PPAR) signaling pathway, and steroid biosynthesis were enriched (Figure 2B). Interestingly, we found that the expressions of key enzymes in fatty acid metabolism such as fatty acid synthase (*fasn)* and salicylate decarboxylase (*scd)* [31], were downregulated in the OF-fed group. We also found lysyl oxidase like 2 (loxl2), a gene that involved in liver fibrosis [32] as well as the sterol O-acyltransferase 1 (*soat1*), a potential biomarker for hepatocellular carcinoma [33], were also downregulated by OF. The selected genes that were downregulated in OF-fed groups are shown in Figure 2C. 

Nonalcoholic fatty liver disease (NAFLD) affects one-third of the population worldwide [34]. Nonalcoholic steatohepatitis (NASH) is a progressive type of NAFLD that can lead to cirrhosis and liver cancer [35]. Therefore, we are very interested to know whether the downregulation of lipogenesis enzyme in OF-fed wild-type fish could also reduce the gene expression that is involved in NAFLD, NASH, cirrhosis, and HCC. Therefore, we would like to verify these results by using qPCR.

Our qPCR results indicated that *mxf*, *hamp*, and *irak3* were upregulated by OF (Figure 3), and five hepatocellular carcinoma-associated genes (*fasn*, *scd*, *loxl2a*, *foxo3b* and *soat1*) were downregulated by OF (Figure 4). The results in Figure 3 echo the findings of Figure 1B, and the gene expression in Figure 4 echo the findings of Figure 2B. Together, we found that OF consumption increased the anti-viral and anti-microbial genes, enhanced the immunity-related genes, and downregulated the expression of lipogenesis, fibrosis, and HCC-related genes in the zebrafish model. These alterations hinted that OF might offer a cytoprotective effect; thus, we next utilized normal hepatocytes cell lines for the mechanism study.

### 3.2. Oligo-Fucoidan Enhances the Viability of Normal Hepatocytes

Clone 9, a rat normal hepatocyte line, was treated with oligo-fucoidan to examine whether oligo-fucoidan offers a cytoprotective effect. Intriguingly, we found that oligo-fucoidan significantly increased the viability of normal liver cells (Figure 5A). Healthy hepatocytes contain abundant Cyp3a4, which is a critical enzyme for oxidizing toxins and drugs mainly in the liver and intestines, whose activity displays a decreasing trend with nonalcoholic fatty liver disease (NAFLD) progression [36,37]. We found that oligo-fucoidan treatment enhanced the Cyp3a4 activity in Clone 9 cells (Figure 5B). We further investigated the upstream regulator of Cyp3a4, the *Hnf4a*, which is a critical nuclear transcriptional factor that is associated with improved liver function: a previous study demonstrated that Hnf4a expression is decreased in hepatic cirrhosis, but the overexpression of Hnf4a resets the transcription factor network in hepatocytes and prevents hepatic failure [38]. We found that oligo-fucoidan significantly upregulated the expression of *Hnf4a* in Clone 9 cells (Figure 5C).

### 3.3. Oligo-Fucoidan Binds to C-Type Lectin-Like Receptor 2 (CLEC-2)-ASGPR1/2 in Hepatocytes

Oligo-Fucoidan exhibits many bioactivities including anti-inflammation, anti-angiogenic, and anti-cancer. However, the receptor for OF remains unknown. Oligo-Fucoidan is a novel platelet agonist for C-type lectin-like receptor 2 (CLEC-2) [7]. The hepatocyte-specific CLEC-2 receptor is an asialoglycoprotein receptor (ASGPR), which is a transmembrane hetero-oligomeric glycoprotein complex composed of two ASGPR1 subunits and one ASGPR2 subunit [8]. We hypothesized that OF binds to the hepatocyte lectin receptor ASGPR1/2 and promotes the expression of *HNF4A* through the JAK2/STAT3 axes. Then, *HNF4A* resets transcription networks and prevents hepatic failure. To prove this hypothesis, we first examined whether OF binds to ASGPR1/2 in hepatocytes in vitro and in vivo. In OF-treated Clone 9 cells, we found that OF reduced the binding capacity of fluorescein isothiocyanate (FITC)-labeled asialofetuin (an ASGPR-binding protein) in liver cells (Figure 6A,B). Using the in vivo model, mouse tail veins were injected with an isotope labeled ^68^Ga-NOTA-HL (galactose binding to ASGPR) together with OF for an in vivo competition assay. Our results indicated that OF reduced the binding capacity in ^68^Ga-NOTA-HL PET imaging (Figure 6C, D). These data suggest that oligo-fucoidan binds to highly conserved hepatic ASGPR in normal livers.

### 3.4. Oligo-Fucoidan Activates Phosphorylation of STAT3 and Increases Hnf4a Protein Levels in Hepatocytes

The ASGPR on hepatocytes has been linked to the activation of the JAK2–STAT3 signaling cascades in response to inflammatory inducing platelets de-sialylation [9]. We hypothesized that oligo-fucoidan binds to ASGPR and activates the JAK2/STAT3 signaling pathway and enhances the nuclear translocation of pSTAT3, thereby increasing *Hnf4a* mRNA expression. Indeed, we found that OF treatment increased pSTAT3 and Hnf4a protein expression in Clone 9 cells (Figure 7).

### 3.5. Oligo-Fucoidan Increases the Expression of P1-Hnf4a in Hepatocytes

There are two isoforms of hepatocyte nuclear factor-4α (HNF4A), the P1- and P2-isoform. The *P1-HNF4A* exhibits a tumor-suppressive effect, and *P2-HNF4A* acts as an oncogene in hepatocyte [39]. Beta-catenin repressed the activity of *P1-Hnf4a* expression and has been reported to be responsible for colon tumorigenesis [40]. We examined whether the P1 isoform was enhanced by OF treatment. Our results indicated that OF treatment promoted the *P1-Hnf4a* expression, but it suppressed the *P2-Hnf4a* dramatically in Clone 9 cells (Figure 8A). The expression level of Asgpr2 was also increased by OF treatment in the 10 μg/mL OF-treated group but there were no changes in the 100 μg/mL OF treatment group (Figure 8B). To explore the role of *Hnf4a* isoforms in hepatocytes, the chemical carcinogen thioacetamide (TAA) was utilized to treat Clone 9 cells. We found that TAA exhibited an opposite effect to hepatocytes compared to OF. TAA treatment downregulated *P1-Hnf4a* but upregulated *P2-Hnf4a* expression in Clone 9 cells (Figure 8C). Thioacetamide also reduced the *Asgpr1* and *Asgpr2* expression levels (Figure 8D).

### 3.6. STAT3 Inhibitor Decreases the Viability of Normal Liver Cells

Since we assume that OF mediates *Hnf4a* through the activation of STAT3, hence, we further applied a STAT3 inhibitor, BBI608, to investigate the hypothesis. Our data revealed that BBI608 decreased either the viability or the Cyp3a4 activity in Clone 9 cells (Figure 9 and B). We further confirmed that BBI608 is capable of suppressing the expression of *Hnf4a* (Figure 9C). This data suggest that the expression of *Hnf4a* was regulated by STAT3 in Clone 9 cells.

### 3.7. The Enhancing Effect of Oligo-Fucoidan on CYP3A4 Activity and Cell Viability Could Be Neutralized by the Knockdown of Hnf4a and Asgpr1 in Normal Liver Cells

We hypothesized Hnf4a contributes to the cyto-protection effect of oligo-fucoidan; thus, we knocked down *Hnf4a* in Clone 9 cells. We found that the enhancement of Cyp3a4 activity promoted by OF was neutralized in Clone 9-shHnf4a (Figure 10A), suggesting that the cyto-protection effect of OF is through activating Hnf4a. Moreover, we knocked down *Asgpr1* and found that the OF-promoted effect in cell viability was eliminated in Clone 9-shAsgpr1 (Figure 10B). Together, our data confirm that the hepatocyte-protective effect of oligo-fucoidan is through the ASGPR/STAT3/HNF4A axis.

## 4. Discussion

Oligo-fucoidan has been demonstrated to have several beneficial effects, including anti-obesity, anti-inflammatory, and anti-tumorigenic effects [41,42,43,44,45]. The oligo-fucoidan used in this study has a lower molecular weight (approximately 667 Da), which shows 40% superior effects and bioactivity compared to polymerized fucoidan. The lower molecular weight might be a critical factor that could further affect the biocompatibility or therapeutic effects, especially at the gene, protein, or even cellular levels. We demonstrate the hepatocyte-protective effect in oligo-fucoidan fed zebrafish, which upregulates immune development-related genes and anti-viral proteins and suppresses lipogenesis and fibrosis genes in a zebrafish model. In addition, oligo-fucoidan enhances cell viability and CYP3A4 activity in normal liver cells. We uncover the potential mechanism of oligo-fucoidan in liver protection, possibly through the ASGPR–STAT3–HNF4A axis. Further investigations of oligo-fucoidan dosages is needed in the clinical practices due to the divergence oligo-fucoidan sources, and various dosages have been used in different demands [43]. We suggest that there will be many benefits and much understanding gained from studying the proper use of oligo-fucoidan.

We used four-month-old wild-type zebrafish to feed oligo-fucoidan for a one-month period by oral gavaging and then sacrificed the fish to collect liver specimens for RNA extraction and microarray analyses. Afterwards, we performed two independent batches at different time courses and compared the differential genes. We found only 124 upregulated genes, and 114 downregulated genes were overlapped between two batches, the reasons could be that oligo-fucoidan has many bioactivities, from anti-inflammation, anti-oxidant, immunoregulation, anti-viral, and anti-coagulant [46,47]. We compared two batches of wild-type fish treated with oligo-fucoidan to find out the most common differentially regulated genes and identified that genes involved in immune development revealed the most significant differences in response to oligo-fucoidan-fed condition. Using zebrafish gene expression profiling, we can obtain the systematic information regarding upregulated or downregulated genes in liver between oligo-fucoidan-fed and wild-type zebrafish. The liver was composed of several cell lineages, 60% of cells in the liver are hepatocytes, and the remaining 40% are non-parenchymal cells (NPCs), liver sinusoidal endothelial cells (LSECs), resident macrophage Kupffer cells (KCs), hepatic stellate cells (HSCs), cholangiocytes, and diverse immune cell types. The current study used Clone 9 normal hepatocytes as *in vitro* model; the alterations in the gene expression profiling using intact fish liver might be different from those of pure hepatocytes if those changes occurred in non-hepatocyte cells. It has been revealed that NASH pathogenesis is linked to the cell-type-specific reprogramming of the liver cell transcriptomes [48]. Therefore, we revealed that oligo-fucoidan enhancements to the immune system might benefit from the changing of Kupffer cells and monocyte-derived macrophages in the liver.

The hepatocyte-protective effect of oligo-fucoidan is an interesting finding in this study. Oligo-fucoidan is considered as a food supplement without potential toxicity. In our experiment, we have used oligo-fucoidan to feed zebrafish, and we have provided evidence that oligo-fucoidan not only boosts the immune system and enhance anti-viral-associated gene expression but also declines lipogenic enzymes and liver fibrosis-related genes. In fact, brown seaweeds containing fucoidan are widely consumed as part of a normal diet in East Asia, particularly in Japan, Korea, Taiwan, and mainland China. However, liver cancer incidence is relatively higher in these countries/regions, which is possibly due to the insufficient regular consumption to reach the effective dose. The main reason for the manufacturing of oligo-fucoidan specifically is because consuming brown seaweed can only ingest very limited amounts of fucoidan. Approximately 6 kg of seaweed extraction can only refine to 1 g of oligo-fucoidan. However, human patients need to consume 8 g of oligo-fucoidan per day, and it is obviously impossible for a patient to consume 48 kg daily. Hence, although the consumption of brown seaweed in Eastern Asia is common in daily diets, there are some limitations and concerns for brown seaweed consumption; namely, brown seaweed contains an abundant amount of iodide, which might be another restriction for cancer patients or those with thyroid or renal diseases. Secondly, different species of seaweed exhibit different compositions of fucoidan (different molecular weight or polymer form) that may have different efficacies of therapeutic effects.

Oligo-fucoidan is provided by Hi-Q Marine Biotech that is not only highly concentrated, but also smaller in molecular weight, enabling efficient uptake for cells. In our study, we have demonstrated that oligo-fucoidan is capable of enhancing the immune system and increasing the amount of anti-viral associated genes in hepatocytes. In addition, oligo-fucoidan suppressed the expression of lipogenic enzymes and liver fibrosis-related genes. A putative intrinsic signaling cascade ASGPR/STAT3/HNF4A has also been investigated via qPCR and PET imaging. Hence, we propose that oligo-fucoidan might have the potential for protecting hepatocyte in zebrafish.

Fucoidan exhibits an anti-cancer effect through the inhibition of PI3K/AKT and MAPK, also activation of the caspase pathways, and interaction with vascular endothelial growth factor (VEGF), bone morphogenetic protein (BMP), transforming growth factor beta (TGF-β), and estrogen receptors [49]. In this study, the GeneTitan assay implied that the driving force of this axis is HNF4A. In a previous study, the role of HNF4A in rodents was reported to be a critical factor to prevent diethylnitrosamine-induced HCC [50]. Our data found that oligo-fucoidan activates HNF4A via the P1 promoter, suggesting that oligo-fucoidan is capable of modulating HNF4A expression through the P1 promoter. This might be the key step of the pathway because of the divergent results of HNF4A when bound to different promoters. We have demonstrated the hepatocyte-protective effect of oligo-fucoidan in WT zebrafish. We propose that the most significant effect of oligo-fucoidan is the prevention of liver damage. Our data strongly support the beneficial effects of oligo-fucoidan with a boosted immune system, fibrosis prevention, lipogenesis suppression, and even attenuating the tumor progression of HCC. Oligo-fucoidan-induced *P1-HNF4A* expression through the phosphorylation of STAT3 elucidates the potential mechanisms that differentially induce cell growth in normal livers and suppress hepatocarcinogenesis. Moreover, from the gene expression, we can understand other possible mechanisms behind the beneficial effects of oligo-fucoidan treatment.

The STAT3 has both tumor-suppressing and tumor-promoting properties, constituting an active version of Stat3alpha that suppresses transformation in mouse embryonic fibroblast [51]. The significant finding of our study is that oligo-fucoidan can bind to the ASGPR1/2 receptor to activate pSTAT3, which can increase the expression of the tumor suppressor P1-HNF4A. Another anti-inflammation and anti-tumor gene, heme oxygenase-1 (HO-1) is induced by Interleukin 6 (IL-6) through the IL-6/Janus kinase (JAK)/STAT3 pathways in hepatoma cells [52]. In addition to STAT3, there are other possible upstream regulators in mediating the expression of HNF4A. In the adult liver, HNF6/Onecut 1(OC1), HNF1B, and HNF1A regulate the expression of HNF4A [53]. However, none of them has connected to the fucoidan activation.

In this study, we have proven that oligo-fucoidan binds to ASGPR1/2 receptors on hepatocytes. This is the first report to reveal the potential receptor for oligo-fucoidan. Our study provides insight into oligo-fucoidan through activation of the ASGPR1/2/STAT3/P1-HNF4A axis, which results in resetting the genetic regulatory networks of hepatocytes and enhancing the overall healthy physiological status of the cells. We suggest that oligo-fucoidan not only protects hepatocytes against microbial and virus infections, but also increases hepatocyte viability and activity. Oligo-fucoidan could be a robust potential candidate for future drug development in liver diseases and even virus infections. 

## 5. Conclusions

Using transcriptomic analysis of the liver from wild-type zebrafish after feeding oligo-fucoidan, we found that genes involved in immune system development were upregulated, especially in that we verified the expression of anti-viral, anti-microbial, and immune modulation proteins by qPCR. Our data suggest that oligo-fucoidan might enhance the anti-viral ability and increase the immunity against virus infection. We also found that genes involved in lipid metabolism, fibrosis, and liver cancer were downregulated, and especially we verified the expression of *fasn*, *scd*, *loxl2a*, *foxo3b,* and *soat1* by qPCR. We have proven that oligo-fucoidan binds to the ASGPR1/2 receptors on hepatocytes, and this is the first report to identify the potential receptor for oligo-fucoidan. We also provide strong evidence that the signal transduction cascade is activated after oligo-fucoidan binds to the ASGPR1/2 receptors, such as a STAT3-mediated signaling cascade. Our study provides insight into oligo-fucoidan through activation of the ASGPR1/2/STAT3/P1-HNF4A axis, which results in resetting the genetic regulatory networks of hepatocytes and enhancing the overall healthy physiological status of the cells. 

## Figures and Tables

**Figure 1 biomolecules-10-00898-f001:**
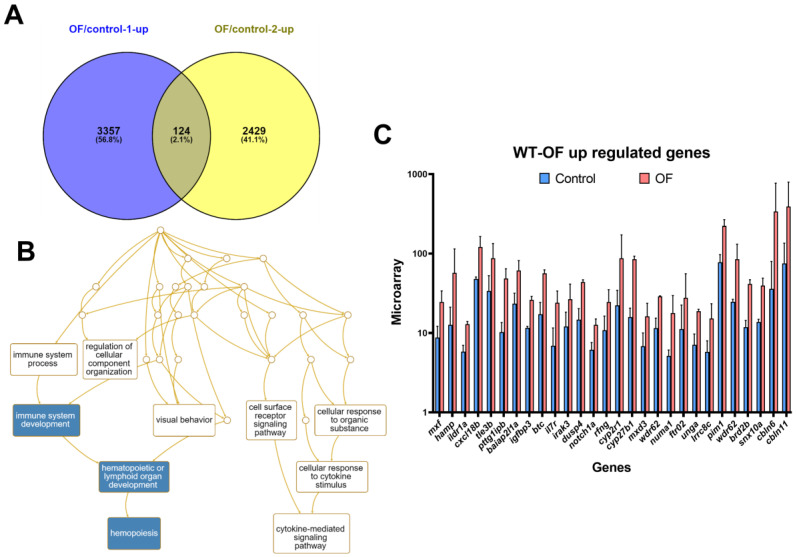
Upregulation of immune system development-associated genes, such as an anti-viral gatekeeper for *mxf* as well as host defense multiple microbial, *hamp,* in oligo-fucoidan (OF)-fed wild-type zebrafish. (**A**) Venn diagram of the upregulated genes from two batches of experiment between OF-fed and control fish; (**B**) Gene ontology analysis revealed genes involved in immune system development, hematopoietic or lymphoid development, and hematopoiesis were enriched. (**C**) The selected genes that are upregulated by OF from the microarray analysis.

**Figure 2 biomolecules-10-00898-f002:**
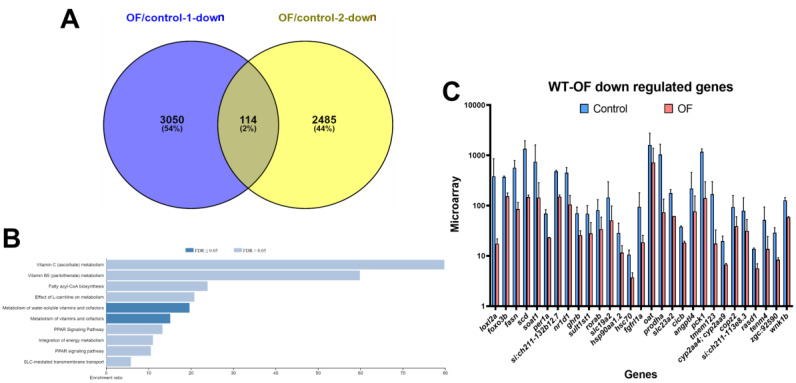
Downregulation of fatty acid metabolism, the peroxisome proliferator-activated receptors (PPAR) signaling pathway, and steroid biosynthesis-associated genes in OF-fed wild-type zebrafish. (**A**) Venn diagram of the downregulated genes from two batches of experiment between OF-fed and control fish. (**B**) Kyoto Encyclopedia of Genes and Genomes (KEGG) pathway analysis revealed that the genes involved in fatty acid metabolism were enriched. (**C**) The selected genes that are downregulated by OF from the microarray analysis.

**Figure 3 biomolecules-10-00898-f003:**
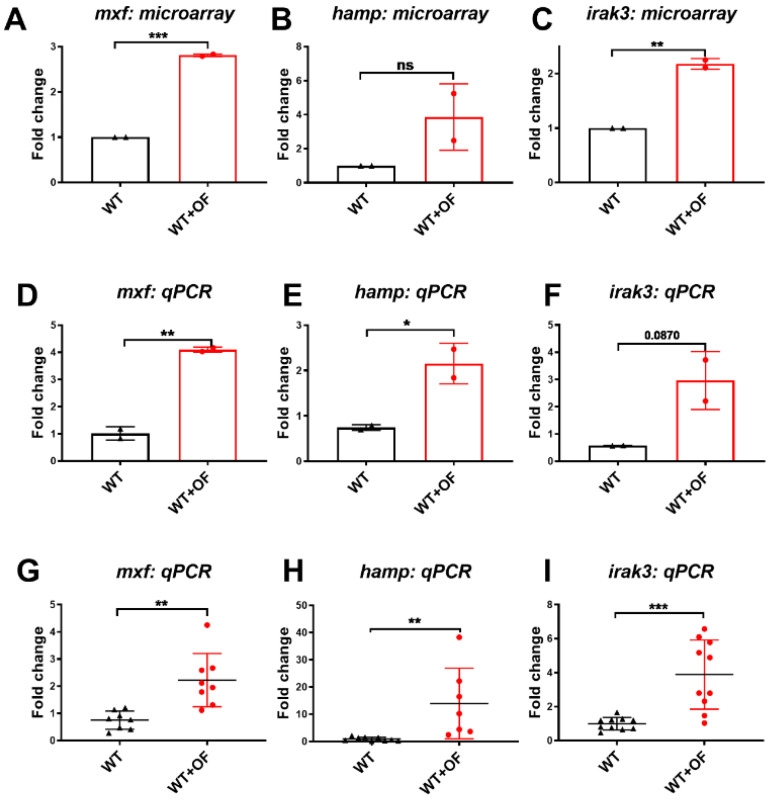
Microarray versus qPCR validation for three upregulated genes in OF-fed and control zebrafish. (**A**–**C**) The fold changes of oligo-fucoidan fed wild-type (WT+OF) zebrafish versus the no-treatment control (WT), where each dot represents the average of two fish from the same batch. (**D**–**F**) Quantitative polymerase chain reaction (qPCR) analyses for the same samples that were used in microarray, where each dot represents average of two fish from the same batch. (**G**–**I**) Quantitative polymerase chain reaction (qPCR) analyses for the all OF-fed wild-type fish (red) versus the untreated controls (black), where each dot represents one fish. Statistical analyses were performed with Student’s t-test (*0.01 < *p* ≤ 0.05; **0.001 < *p* ≤ 0.01; *** ≤ 0.001, ns: non-significance).

**Figure 4 biomolecules-10-00898-f004:**
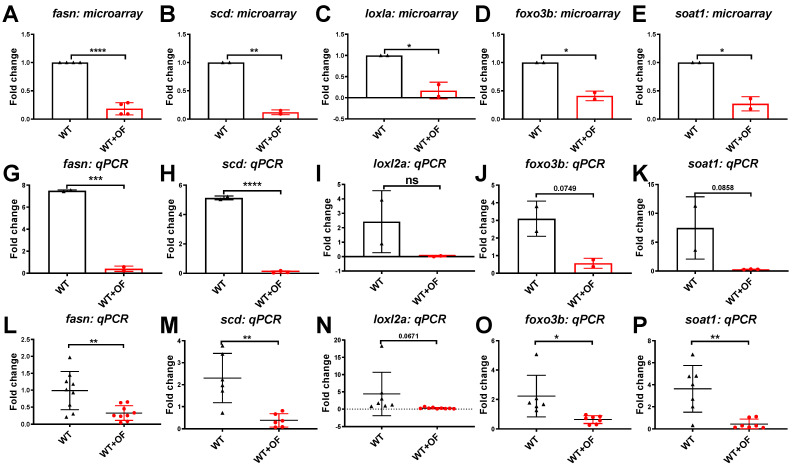
Microarray versus qPCR validation for five downregulated genes (*fasn*, *scd*, *loxl2a*, *foxo3b* and *soat1*) in OF-fed and control zebrafish. (**A**–**E**) The fold changes of OF-fed wild-type zebrafish versus the no treatment controls, where each dot represents the average of two fish. (**G**–**K**) qPCR for the same samples that were used in the microarray, where each dot represents the average of two fish. (**L**–**P**) qPCR analyses for all OF-fed fish (red) versus untreated controls (black), where each dot represents one fish. Statistical analyses were performed with student’s t-test (* 0.01 < *p* ≤ 0.05; ** 0.001 < *p* ≤ 0.01; *** 0.0001 < *p* ≤ 0.001; **** *p* ≤ 0.0001).

**Figure 5 biomolecules-10-00898-f005:**
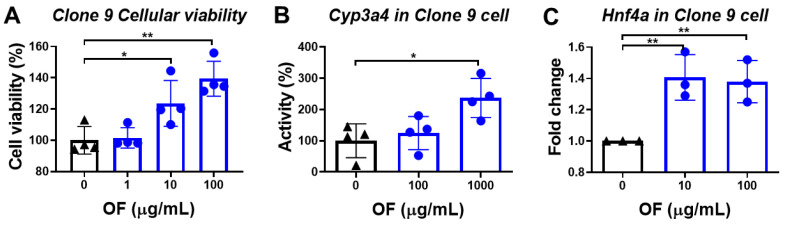
Oligo-fucoidan (OF) enhanced the viability and increased the Cyp3a4 activity in normal liver Clone 9 cells. (**A**) Oligo-fucoidan enhanced the viability of normal liver cells. The black color indicated control cells without oligo-fucoidan treatment, the blue color denoted oligo-fucoidan treatment at different concentration. The cellular viability was measured by WST-1 assay, and the cell viability (%) was compared to control cells. (**B**) Cyp3a4 activity in Clone 9 cells was increased by treatment with oligo-fucoidan. The black color indicated control cells without oligo-fucoidan treatment, while the blue color denoted oligo-fucoidan treatment at different concentrations. The Cyp3a4 activities of the oligo-fucoidan-treated cells or control cells were measured by a CYP3A4 activity assay Kit assay, and the activity (%) was compared to the average of activity of the control cells. (**C**) The mRNA expression level of Hnf4a was increased by treatment with oligo-fucoidan. The black color indicated control cells without oligo-fucoidan treatment, while the blue color denoted oligo-fucoidan treatment at different concentrations. The expression of Hnf4a was measured by qPCR, and the fold changes were compared to control cells. *0.01 < *p* ≤ 0.05; ***p* ≤ 0.01.

**Figure 6 biomolecules-10-00898-f006:**
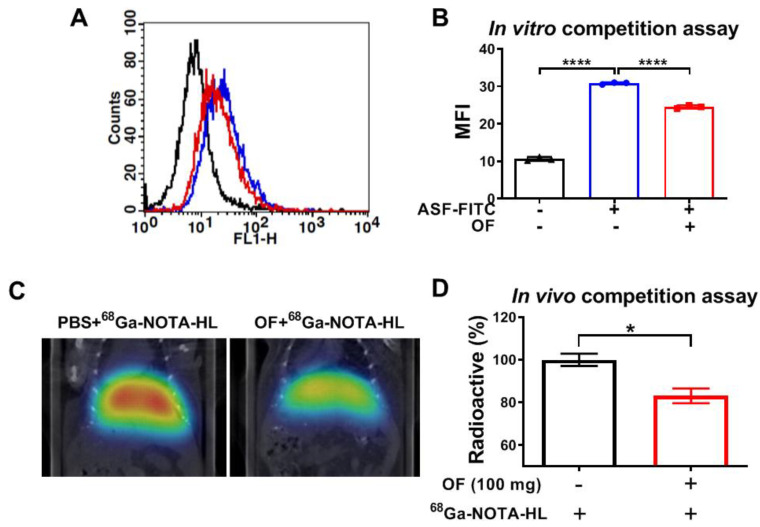
In vitro and in vivo competition assays demonstrated that oligo-fucoidan binds to C-type lectin-like receptor 2 (CLEC-2)–asialoglycoprotein receptors (ASGPR)1/2 in hepatocytes. (**A**) Flow cytometry profiles of Clone 9 cells. The black line histogram represents isotype control, the blue indicates cells stained by asialofetuin–fluorescein isothiocyanate (ASF-FITC), and the red line denotes cells treated with OF and stained by asialofetuin–FITC (ASF-FITC). (**B**) The mean fluorescence intensity (MFI) profiles of Clone 9 cells with or without OF treatment in the flow cytometry analysis. The black color represents isotype control, blue indicates cells with asialofetuin–FITC (ASF-FITC), and the red line denotes cells treated with OF and asialofetuin–FITC (ASF-FITC). (**C**) ^68^Ga-NOTA-hexavalent lactoside (HL) positron emission tomography (PET) imaging of the murine livers demonstrates the in vivo ASGPR binding image *in vivo*. Phosphate-buffered saline (PBS) and OF (100 mg) were administrated through an intravenous injection with ^68^Ga-NOTA-HL (^68^Ga-2,2’,2’’-(1,4,7-triazonane-1,4,7-triyl) triacetic acid-hexavalent lactoside) injection (*N* = 3/group) simultaneously. (**D**) The relative radioactive intensity of ^68^Ga-NOTA-HL in vivo between the PBS and OF-injected groups. The bars indicate the mean ± SD, where the black bar represents the PBS group and the red bar indicates the OF treatment group. The radioactivity (%) was compared to PBS (no fucoidan treatment) group. Statistical significance was calculated by Student’s t-test (**p* ≤ 0.05; ***p* ≤ 0.01; ****p* ≤ 0.001; *****p* ≤ 0.0001).

**Figure 7 biomolecules-10-00898-f007:**
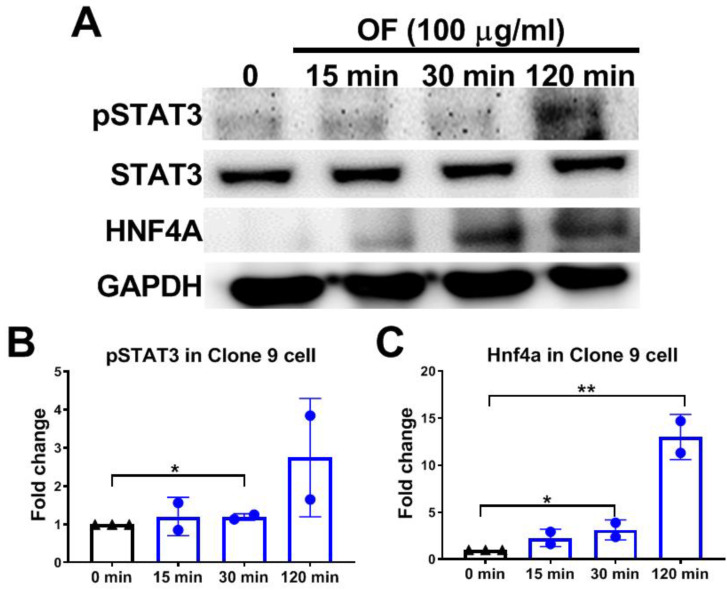
OF enhanced the phosphorylation of signal transducer and activator of transcription 3 (pSTAT3) and increased hepatocyte nuclear factor 4 alpha (Hnf4a) protein levels. (**A**) Immunoblotting of pSTAT3, STAT3, and Hnf4a of Clone 9 cells treated with oligo-fucoidan (100 µg/ml) for 0, 15, 30, and 120 min. GAPDH was used as internal control. (**B**) Statistical analysis of pSTAT3 staining in three batches of oligo-fucoidan-treated Clone 9 cells. The back color indicated no treatment control, while the blue color denoted oligo-fucoidan treatment for 15, 30, or 120 min as indicated at the X-axis. The pSTAT3 intensity was quantified from Western blotting, and the fold changes were compared to control cells (0 min group). (**C**) Statistical analysis of Hnf4a levels in three batches of oligo-fucoidan-treated Clone 9 cells. The back color indicated no treatment control, the blue color denoted oligo-fucoidan treatment for 15, 30, or 120 min, as indicated at the *X*-axis. The Hnf4a level was quantified from Western blotting, and the fold changes were compared to control cells (0 min group). Statistical significance was calculated by Student’s t-test (**p* ≤ 0.05; ***p* ≤ 0.01).

**Figure 8 biomolecules-10-00898-f008:**
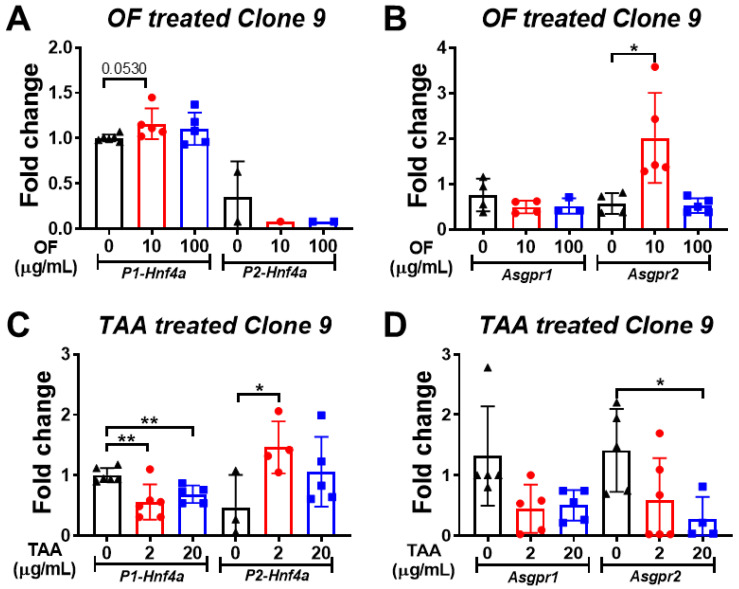
Regulations of P1-Hnf4a and Asgpr1/2 expression in normal hepatocytes by oligo-fucoidan (OF) and carcinogen (thioacetamide, TAA). (**A**) Expression profiles of the P1 and P2 isoforms of Hnf4a after being treated with different concentrations of OF in Clone 9 hepatocytes. The black color indicated control cells without oligo-fucoidan treatment, the red color denoted oligo-fucoidan at 10 µg/mL, and blue color represented the oligo-fucoidan at 100 µg/mL concentration. The expression of P1-Hnf4a or P2-Hnf4a (indicated below the bar) was measured by qPCR, and the fold changes were compared to control cells. (**B**) Gene expression profiles of ASGPR1/2 after different concentrations of OF in Clone 9 hepatocytes. The black color indicated control cells without oligo-fucoidan treatment, the red color denoted oligo-fucoidan at 10 µg/mL, and the blue color represented the oligo-fucoidan at 100 µg/mL concentration. The expression of Asgpr1 or Asgpr2 (indicated below the bar) was measured by qPCR, and the fold change was compared to control cells. (**C**) Expression profiles of the P1 and P2 isoforms of Hnf4a after being treated with different concentrations of thioacetamide (TAA) in clone 9 hepatocytes. The black color indicated control cells without oligo-fucoidan treatment, the red color denoted TAA at 2 µg/mL, and blue color represented the oligo-fucoidan at 20 µg/mL concentration. The expression of P1-Hnf4a or P2-Hnf4a (indicated below the bar) were measured by qPCR, and the fold changes were compared to control cells. (**D**) Gene expression profiles of Asgpr1/2 after different concentrations of TAA in Clone 9 hepatocytes. The black color indicated control cells without oligo-fucoidan treatment, the red color denoted TAA at 2 µg/mL, and the blue color represented the TAA at 20 µg/mL concentration. The expression of Asgpr1 or Asgpr2 (indicated below the bar) was measured by qPCR, and the fold changes were compared to control cells. Statistical significance was calculated by t-test (**p* ≤ 0.05; ***p* ≤ 0.01).

**Figure 9 biomolecules-10-00898-f009:**
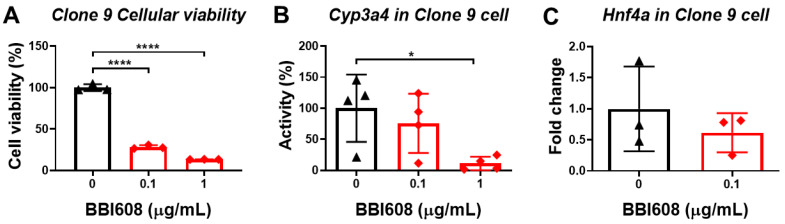
STAT3 regulated *Hnf4a* expression in Clone 9 cells. (**A**) STAT3 inhibitor (BBI608) decreased the viability of normal liver cells. The black color indicated control cells without STAT3 inhibitor treatment, the red color denoted BBI608 treatment at different concentrations. The cellular viability was measured by WST-1 assay, and the cell viability (%) was compared to control cells. (**B**) Cyp3a4 activity in Clone 9 cells was decreased by treatment with a STAT3 inhibitor (BBI608). The black color indicated control cells without STAT3 inhibitor treatment, and the red color denoted BBI608 treatment at different concentration. The Cyp3a4 activities of the STAT3 inhibitor-treated cells or control cells were measured by CYP3A4 Activity Assay Kit assay, and the activity (%) was compared to the average of activity of the control cells. (**C**) The mRNA expression level of Hnf4a was decreased by treatment with a STAT3 inhibitor (BBI608). The black color indicated control cells without STAT3 inhibitor treatment, and the red color denoted BBI608 treatment at 0.1 µg/ml. The expression of Hnf4a was measured by qPCR, and the fold changes were compared to control cells. Statistical significance was calculated by Student’s t-test (*0.01 < *p* ≤ 0.05; **0.001 < *p* ≤ 0.01; ***0.0001 < *p* ≤ 0.001; *****p* ≤ 0.0001).

**Figure 10 biomolecules-10-00898-f010:**
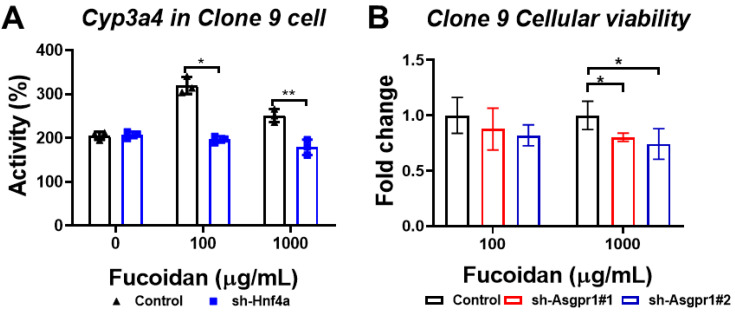
Knockdown of Hnf4a and Asgpr1 in Clone 9 cells neutralized the OF-promoting effects in both Cyp3a4 activity and the cellular viability of normal liver cells. (**A**) Knockdown of Hnf4a abolished the fucoidan-mediated enhancement of Cyp3a4 activity. The blue color indicated sh-Hnf4a transfected cells, and the black color denoted the control transfected cells. The Cyp3a4 activities of the sh-Hnf4a knockdown or control cells were measured by CYP3A4 Activity Assay Kit. (**B**) The knockdown of Asgpr1 abolished the fucoidan-mediated enhancement of cellular viability in normal liver cells. The red color indicated first short hairpin RNA (shRNA) against Asgpr1-transfected cells, and the blue color denoted the second sh-RNA against Asgpr1-transfected cells, and the black color showed the control-transfected cells. The cellular viability was determined using WST-1 assay, and the fold changes were compared to control cells. Statistical significance was calculated by Student’s t-test (**p* ≤ 0.05; ***p* ≤ 0.01).

**Table 1 biomolecules-10-00898-t001:** Primer sequences.

Primer Name	sequence	Accession Number
mxf-F	CGACTGGGGAGGATGTTAAA	NM_182942.4
mxf-R	CCCCGGTACTTGACTTCGTA	
hamp-F	CAGCAGGTACAGGATGAGCA	NM_205583.2
hamp-R	AGCCTTTATTGCGACAGCAT	
irak3-F	GGCATTTCCACCAGGACTTA	XM_017355295.2
irak3-R	AGAAGCGATCCGAATCTGAA	
fasn-F	ATCTGTTCCTGTTCGATGGC	XM_682295
fasn-Rscd-Fscd-Rloxl2a-Floxl2a-Rfoxo3b-Ffoxo3b-Rsoat1-Fsoat1-R	AGCATATCTCGGCTGACGTTACGCTCCTCAGATACGCACTAGTCGTAGGGAAACGTGTGGCGAATTCCTGGTCACAGGTTTCGGCTGTTTAAAGGTGTCCAGAGAGCACCCCTGACAAGACACGAGCTCTTTCCAGTTCCGCTCTCATAAGGTGGCTTCGGCCTCATCAGGTCTGCTTTC	NM_198815XM_009305250.3NM_131085.1NM_001123272.1
actin-F	CTCCATCATGAAGTGCGACGT	NM_131031.1
actin-F	CAGACGGAGTATTTGCGCTCA	
P1-Hnf4α-Q-F	CATGGATATGGCCGACTACAGCGC	NM_008261.3
Hnf4α-Q-R	GCCCGAATGTCGCCATTGATCC	
P2-Hnf4α-Q-F	CAGGGGTTCCTGCAGATCACATCA	NM_001312907.1
Hnf4α-Q-R	GCCCGAATGTCGCCATTGATCC	
Asgr1-F	ATCCCAAAATTCCCAACTCC	XM_017314231.1
Asgr1-R	TTTCCAGCTTCGACTCCACT	
Asgr2-F	CCTGTTGGTGGTCATCTGTG	NM_001313925.1
Asgr2-R	GGTCTGCCTTTAGCTGTTGC

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
