# Peer review of "Transcriptomically Revealed Oligo-Fucoidan Enhances the Immune System and Protects Hepatocytes via the ASGPR/STAT3/HNF4A Axis"

_biomolecules, 2020, doi:10.3390/biom10060898_

Round 1

Reviewer 1 Report

In this paper, Cheng et al. indicated that oligo-fucoidan decreasing fatty liver and liver fibrosis via activation of the ASGPR/STAT3/HNF4A axes. I have some comments shown as following:

  1. Section 2.3: Were zebrafish anesthetized before oral gavage? If yes, please describe the method and indicate the reference. If not, the method of oral gavage for zebrafish should be given.

  1. How many zebrafish were used for Gene Expression Profiling analysis? How do you calculate the statistic difference?

  1. It is not clear the rational why authors detected mxf, hamp, irak3, and fasn genes (Fig. 3 and 4)

  1. Does the gene expression in Fig. 3 echoes to the findings of Fig. 2B.

  1. Fig. 5, whether Cyp3a4 and hnf4a contribute to cyto-protection effect of OF should be tested by inhibitors or gene silencing. Moreover, the exact mechanisms of Cyp3a4 and hnf4a in protection of liver function should also be tested in vitro and in vivo.

  1. HNF4A resets transcription networks and prevents hepatic failure

  1. expression of Hnf4a is decreased in hepatic cirrhosis, but the overexpression of Hnf4a resets the transcription factor network in hepatocytes and prevents hepatic failure

  1. Fig. 7, the inhibitor of ASGPR or Stat3 pathways is needed to confirm the expression of Hnf4a were regulated by ASGPR or Stat3.

  1. Fig. 8B, how do you explain that the expression of ASGPR2 was decreased in response to 100 μg/mL OF?

  1. Fig. 9, would BBI608 reverse cyto-protective effects of OF when combined with OF? Similarly, would BBI608 change the expression of Cyp3a4 and Hnf4a if combined with OF?

  1. Fig 10, did the expression of Stat3 and Hnf4a change after knockdown of ASGPR2?
  2. Discussion: Since Stat3 is well recognized as an oncogene and has effect in promoting inflammation, what is the significance of Stat3 in this paper? In addition to Stat3, what is the possible up-stream regulator in mediating the expression of Hnf4a?

  1. Discussion. Do the result of zebrafish from Gene Expression Profiling echo to the results of in vitro models? If no, the author should discuss the benefit of using zebrafish in this experiment.

  1. Because the author did not use any model of immune system activation, viruses and microbial infection, nor fatty liver and liver fibrosis, therefore, the conclusion (and abstract as well) is not convincible and inadequate.

Author Response

Reviewer #1 Comments: [In this paper, Cheng et al. indicated that oligo-fucoidan decreasing fatty liver and liver fibrosis via activation of the ASGPR/STAT3/HNF4A axes. I have some comments shown as following:

Section 2.3: Were zebrafish anesthetized before oral gavage? If yes, please describe the method and indicate the reference. If not, the method of oral gavage for zebrafish should be given.

RESPONSE: [Thank you for providing these insights. Oral gavaging was conducted as described previously (J Vis Exp. 2013; (78): 50691), the fish were anesthetized using MS-222 solution (150 mg/L), and 5 μl of solution was injected slowly into the intestinal tract of adult fish using flexible tubing. We have rewritten the Section 2.3, provided the method of oral gavage and cite this paper. ]

How many zebrafish were used for Gene Expression Profiling analysis? How do you calculate the statistic difference?

RESPONSE: [ [Thank you for the comments. Each batch, two fish were used for Gene Expression Profiling analysis. Since we have two batches, so totally we have the Gene Expression Profiling from four fish. We calculate the statistic different using t-test. ]

It is not clear the rationales why authors detected mxf, hamp, irak3, and fasn genes (Fig. 3 and 4)

RESPONSE: [Thank you for the comments. The rational to verify the mxf, hamp and irak3 is that currently we are experiencing COVID-19 pandemic, we are very intriguing by the upregulation of anti-viral, anti-microbial and immune modulation proteins by oligo-fucoidan feeding to wild-type fish. If this is true, normal people might be able to take oligo-fucoidan to enhance the anti-viral ability and increase the immunity against COVID-19. The rational to verify the fasn gene is that Nonalcoholic fatty liver disease (NAFLD) is affecting one third of population worldwide, and a serious type of NAFLD is Nonalcoholic steatohepatitis (NASH) which can lead cirrhosis and liver cancer. Therefore, we are very interested to know whether normal fish can decrease the expression of lipogenesis enzyme by taking oligo-fucoidan, it could reduce the NAFLD, NASH, cirrhosis and HCC. I have added those rationales to Section 3.1. ]

Does the gene expression in Fig. 3 echoes to the findings of Fig. 2B.

RESPONSE: [Thank you for the question. The gene expression in Fig. 3 echoes to the findings of Fig. 1B, and the gene expression in Fig. 4 echoes to the findings of Fig. 2B. Using wild-type zebrafish as model, we found oral gavage oligo-fucoidan can increase the anti-viral and anti-microbial proteins, enhance the immunity related genes, and downregulated the expression of genes related to lipogenesis, fibrosis and HCC. I have added those to Section 3.1. ]

Fig. 5, whether Cyp3a4 and hnf4a contribute to cyto-protection effect of OF should be tested by inhibitors or gene silencing. Moreover, the exact mechanisms of Cyp3a4 and hnf4a in protection of liver function should also be tested in vitro and in vivo.

RESPONSE: [Thank you for the comments. Cyp3a4 is an indicator for hepatocyte healthy status as healthy liver contains higher levels of CYP3A4, and NAFLD and diabetes are associated with the decreased hepatic CYP3A4 activity, all of those have been reported previously. We hypothesize HNF4A contribute to cyto-protection effect of oligo-fucoidan, so we knockdown Hnf4a in normal hepatocyte (Clone 9), and found the OF induced increased the Cyp3a4 activity was eliminated (Fig. 10A), and confirmed the cyto-protection effect is regulated by Hnf4a. I have added those statement in the Section 3.7. ]

HNF4A resets transcription networks and prevents hepatic failure. Expression of Hnf4a is decreased in hepatic cirrhosis, but the overexpression of Hnf4a resets the transcription factor network in hepatocytes and prevents hepatic failure.

Fig. 7, the inhibitor of ASGPR or Stat3 pathways is needed to confirm the expression of Hnf4a were regulated by ASGPR or Stat3.

RESPONSE: [Thank you for the constructive suggestion. We knockdown Asgpr1 in normal hepatocyte (Clone 9), and found the OF induced increased the cell viability was eliminated (Fig. 10B). We used STAT3 inhibitor to treat the normal hepatocyte (Clone 9), and found the STAT3 inhibitor decreased the viability and the Cyp3a4 activity of normal liver cells (Fig. 9A, B). Those result indicated Asgpr and Stat3 is involved in the hepatocyte viability. We have used STAT3 inhibitor to treat the normal hepatocyte (Clone 9), and confirmed the expression of Hnf4a is regulated by Stat3 (Fig. 9C). I have added those statement in the Section 3.6. ]

Fig. 8B, how do you explain that the expression of ASGPR2 was decreased in response to 100 μg/mL OF?

RESPONSE: [The expression of ASGPR2 was increased in 10 μg/mL OF, but not difference in 100 μg/mL OF, might due to the over-saturation effect. ]

Fig. 9, would BBI608 reverse cyto-protective effects of OF when combined with OF? Similarly, would BBI608 change the expression of Cyp3a4 and Hnf4a if combined with OF?

RESPONSE: [Thank you for the comments. We found that OF treatment increased the normal hepatocyte viability, Cyp3A4 activity and Hnf4a level (Fig. 5). In the contrast, BBI608 had the opposite effect (Fig. 8). Moreover, we found that knockdown of HNF4A (Fig. 10A) and Asgpr1 (Fig. 10B) neutralized CYP3A4 activity and hepatocyte viability that was mediated by OF treatment. These data confirm the hepatocyte-protective effect of fucoidan. Fucoidan might have other function in hepatocyte. Fucoidan suppresses hepatitis B virus replication by enhancing pERK activation was reported (Virology Journal (2017) 14:178). pERK was increased in 100 µg/ml Fucoidan treatment, and U0126 (ERK inhibitor) treatment could inhibit ERK activation. But, according to the published data, the INF-a mRNA level was partially reversed by ERK inhibitor, indicating there might be other pathways activated by fucoidan. If treated with OF and inhibit the STAT3 with BBI608, we might not observe the fully reverse due to other pathways been activated by OF. ]

Fig 10, did the expression of Stat3 and Hnf4a change after knockdown of ASGPR2?

RESPONSE: [Thank you for the comments. The asialoglycoprotein receptor (ASGPR) has two polypeptide, the major component ASGPR1 and minor component ASGPR2. We used two shRNA to knockdown the major component Asgpr1 in normal hepatocyte, and found they both can reduce OF mediated enhancement of cell viability. We believe Asgpr2 has the same effect since the receptor is composed by Asgpr1/2. ]

Discussion: Since Stat3 is well recognized as an oncogene and has effect in promoting inflammation, what is the significance of Stat3 in this paper? In addition to Stat3, what is the possible up-stream regulator in mediating the expression of Hnf4a?

RESPONSE: [Thank you for the comments. STAT3 has both - tumor suppressing and tumor promoting properties, constitutive active version of Stat3alpha suppress transformation in mouse embryonic fibroblast (Front Biosci,2009,14:2944-58). The significance finding of our study is fucoidan can bind to ASGPR1/2 receptor and activate pSTAT3 which can increase the expression of tumor suppressor P1-HNF4A. So, the effect of pSTAT3 as transcription factor should depend on the genes activated by pSTAT3, so it could function as tumor suppressor in this case. Another anti-inflammation and antitumor gene, Heme oxygenase-1 (HO-1) is the induced by IL-6 through the IL-6/JAK/STAT3 pathways in hepatoma cells (Antioxidants, 2020,9(3):251). In addition to STAT3, there are other possible up-stream regulator in mediating the expression of HNF4A. In the adult liver, HNF6/Onecut 1(OC1), HNF1B and HNF1A regulate the expression of HNF4A (Journal of Hepatology 2018 vol. 68, 1033–1048). However, none of them has connected to the fucoidan activation. I have added those in the discussion section. ]

Discussion. Do the result of zebrafish from Gene Expression Profiling echo to the results of in vitro models? If no, the author should discuss the benefit of using zebrafish in this experiment.

RESPONSE: [Thank you for the question. Using zebrafish gene expression profiling we can obtain the systematic information regarding genes expressing in liver are up- or down-regulated by oligo-fucoidan oral gavage. Because 60% of cells in liver are hepatocytes, but other 40% are others cells including non-parenchymal cells (NPCs), liver sinusoidal endothelial cells (LSECs), resident macrophage Kupffer cells (KCs), hepatic stellate cells (HSCs), cholangiocytes and diverse immune cell types. In vitro model is using normal hepatocyte, the changing from gene expression profiling using liver might be different from hepatocyte if those changing are happened in non-hepatocyte cells. It has been revealed that NASH pathogenesis is linked to cell-type-specific reprogramming of the liver cell transcriptomes (Molecular Cell, 2019, vol. 75 (3), 644-660.e5). Therefore, we revealed oligo-fucoidan enhances the immune system might contributed from the changing of resident macrophage Kupffer cells (KCs) and monocyte-derived macrophages (MDMs) from livers by Oligo-Fucoidan treatment. I have added those in the discussion section. ]

Because the author did not use any model of immune system activation, viruses and microbial infection, nor fatty liver and liver fibrosis, therefore, the conclusion (and abstract as well) is not convincible and inadequate.

RESPONSE: [Thank you for your critical and important comments. I have changed the conclusion as following:

Using transcriptomic analysis the liver of wild-type zebrafish after feeding oligo-fucoidan, we found genes involved in immune system development were upregulated, especially we verified the expression of anti-viral, anti-microbial and immune modulation proteins by qPCR. Our data suggests oligo-fucoidan might enhance the anti-viral ability and increase the immunity against virus infection such as COVID-19. We also found genes involved in lipid metabolism, fibrosis and liver cancer were upregulated, and especially we verified the expression of fasn, scd, loxl2a, foxo3b and soat1 by qPCR. Our data suggests normal people taking oligo-fucoidan might reduce the chances of getting NAFLD, NASH, cirrhosis and HCC. This suggests that taking oligo-fucoidan can prevent liver disease in clinical practice. Using normal hepatocyte as in vitro model, we have proven that oligo-fucoidan binds to the ASGPR1/2 receptors on hepatocytes, and this is the first report on the possible receptor of oligo-fucoidan. We also provide strong evidence that the signal transduction cascade activates after oligo-fucoidan binds to the ASGPR1/2 receptors, such as a STAT3-mediated signaling cascade.

I have changed the abstract as following:

Using an oligo-fucoidan oral gavage in adult wild-type zebrafish, we find that oligo-fucoidan pretreatment enhances the immune system and increases the amount of anti-viral proteins in hepatocytes. Oligo-fucoidan pretreatment also decreases the expression of lipogenic enzymes and liver fibrosis genes. Using pathway analysis, we identify hepatocyte nuclear factor 4 alpha (HNF4A) to be the driver gene. Using normal hepatocyte as in-vitro model, we demonstrate that oligo-fucoidan increase cell viability, Cyp3a4 activity and Hnf4a expression. We also demonstrate that oligo-fucoidan might bind to asialoglycoprotein receptors (ASGPR) in normal hepatocytes through both in vitro and in vivo competition assays. This binding, consequently activating the signal transducer and activator of transcription 3 (STAT3), increases the expression of the P1 isoform of HNF4A, which has a tumor-suppressive activity. Oligo-fucoidan treatment increases the viability of normal liver cells here. Our data suggests oligo-fucoidan might enhance the anti-viral ability and increase the immunity against virus infection, and oligo-fucoidan might reduce NAFLD, NASH, cirrhosis and HCC. This suggests that taking oligo-fucoidan can prevent liver disease in clinical practice. Our experiments demonstrate that oligo-fucoidan activate HNF4A through ASGPR/STAT3 axis.

Reviewer 2 Report

In this manuscript, Cheng et al. investigated the effect of oligo-fucoidan in normal hepatocytes using transcriptomic analysis.

Overall, the work is interesting, but I would like to highlight some aspects that would need to be fixed by the authors:
- First of all, it is not clear why in the era of RNA-seq, microarray technique is used. Which is the advantage in this case of study of using microarrays?
- Even if an English language editing has been declared to be done, I would like to suggest some points that would require an additional editing, such as rows 59-61, rows 189-194, rows 211-212, row 222, rows 257-259, rows 299-301, and rows 360-361.
- In Section 3.1 a deregulation of lipogenesis is mentioned but without giving an in depth explanation of the obtained results
- Figures 3 and 4: "ns" is not defined and the legend is not very clear. Above all, why RT-PCR and qPCR are interchangeable used?
- row 56: "OF" term is mentioned before being explained
- row 206: "p-value" instead of "P-value"

Author Response

Reviewer #2 Comments: [In this manuscript, Cheng et al. investigated the effect of oligo-fucoidan in normal hepatocytes using transcriptomic analysis.

Overall, the work is interesting, but I would like to highlight some aspects that would need to be fixed by the authors:

- First of all, it is not clear why in the era of RNA-seq, microarray technique is used. Which is the advantage in this case of study of using microarrays?

RESPONSE: [Thank you for the question. GeneTitan array is the highest throughput array with multiple probes per gene, results is reproducible, and it comes with software, Transcriptome Analysis Console (TAC), developed by Affymetrix team to analyze the data anytime, and anyway you want, so it is very convenient. Transcriptome Analysis Console (TAC) combines the CEL file analysis and QC features of Expression Console and the statistical analysis of TAC into a single streamlined software application. On the other hand, I have been using RNA-seq, the disadvantage is the analysis need to be done by expert, and set the parameter carefully, otherwise, the differential expression genes were very few. ]

- Even if an English language editing has been declared to be done, I would like to suggest some points that would require an additional editing, such as rows 59-61, rows 189-194, rows 211-212, row 222, rows 257-259, rows 299-301, and rows 360-361.

RESPONSE: [Thank you for your constructive suggestions, I have edit those parts as followed:

rows 59-61 has been edited to

In addition, the galactose exposed by desialylated platelets are recognized by ASGPR in hepatocytes and activates the phosphorylation of STAT3 (pSTAT3) [9], therefore, we assumed that galactose exposed by oligo-fucoidan can bind to ASGPR and activating pSTAT3 in normal hepatocytes to have cyto-protecting effect.

rows 189-194 has been edited to

The gene ontology analysis revealed those upregulated genes were involved in immune system development (Figure 1B). Especially, notch1a, interleukin 7 receptor (il7r), interleukin-1 receptor-associated kinase 3 (irak3), cytochrome P450, family 2, subfamily R, polypeptide 1 (cyp2r1), TLE family member 3, transcriptional corepressor b (tle3b), and PTTG1 interacting protein B (pttg1ipb) were upregulated by OF treatment in the wild-type fish.

rows 211-212 has been edited to

Interestingly, we found that the expressions of key enzymes in fatty acid metabolism, fatty acid synthase (fasn) and salicylate decarboxylase (scd) [21] were downregulated by OF. We also found gene involved with liver fibrosis, lysyl oxidase like 2 (loxl2) [22], was also downregulated by OF.

row 222 has been edited to

After qPCR analysis of the OF treated WT fish, we verified mxf, hamp, and irak3 were upregulated by OF (Figures 3), and five genes (fasn, scd, loxl2a, foxo3b and soat1) were downregulated by OF (Figures 4).

rows 257-259 has been edited to

Fucoidan exhibited many bioactivities including anti-inflammation, antiangiogenic, and anticancer, however, the receptor for fucoidan remains unknown.

rows 299-301 has been edited to

Hepatocyte nuclear factor-4α (HNF4A) exists two isoforms driven by an alternate promoter: P1-HNF4A exhibits tumor-suppressive activity, P2-HNF4A acts as oncogene in hepatocyte.

rows 360-361 has been edited to

Oligo-fucoidan is considered as food supplement and not a drug, therefore limits the usage of oligo-fucoidan in treating patients clinically. ]

- In Section 3.1 a deregulation of lipogenesis is mentioned but without giving an in depth explanation of the obtained results

RESPONSE: [Thank you for your constructive suggestions, I have given an in depth explanation as followed.

Interestingly, we found that the expressions of key enzymes in fatty acid metabolism, fatty acid synthase (fasn) and salicylate decarboxylase (scd) [21] were downregulated by OF. We also found gene involved with liver fibrosis, lysyl oxidase like 2 (loxl2) [22], was also downregulated by OF. Moreover, we found sterol O-acyltransferase 1 (soat1), which has been identified as a potential biomarker for hepatocellular carcinoma [23], was also downregulated by OF. Nonalcoholic fatty liver disease (NAFLD) is affecting one third of population worldwide [24], and a serious type of NAFLD is Nonalcoholic steatohepatitis (NASH) which can lead cirrhosis and liver cancer [25]. Therefore, we are very interested to know whether normal fish can decrease the expression of lipogenesis enzyme by taking oligo-fucoidan, it could reduce the NAFLD, NASH, cirrhosis and HCC. So we would like to verify this results using qPCR.

After qPCR analysis of the OF treated WT fish, we verified mxf, hamp, and irak3 were upregulated by OF (Figures 3), and five genes (fasn, scd, loxl2a, foxo3b and soat1) were downregulated by OF (Figures 4). The gene expression in Fig. 3 echoes to the findings of Fig. 1B, and the gene expression in Fig. 4 echoes to the findings of Fig. 2B. Using wild-type zebrafish as model, we found oral gavage oligo-fucoidan can increase the anti-viral and anti-microbial proteins, enhance the immunity related genes, and downregulated the expression of genes related to lipogenesis, fibrosis and HCC. ]

- Figures 3 and 4: "ns" is not defined and the legend is not very clear. Above all, why RT-PCR and qPCR are interchangeable used?

RESPONSE: [Thank you for your comment, I have defined the “ns” as “non-specific” and edited RT-PCR to qPCR for consistent, and edited the legend clearer. ]

- row 56: "OF" term is mentioned before being explained

RESPONSE: [Thank you for your comment, I have explained the OF as oligo-fucoidan and put OF in parentheses. ]

- row 206: "p-value" instead of "P-value"

RESPONSE: [Thank you for your comment, I have change the “P-value” to “p-value”. ]

Reviewer 3 Report

In the presented study, Cheng et al. investigated the effect of oligo-fucoidan (OF) (extracted from brown seaweed) in hepatocytes. The authors used zebrafish treated with oligo-fucoidan and perform transcriptomics study. The differentially expressed genes identified from transcriptomics data were further confirmed with the qRT PCR. Next, using network analysis, the authors had picked hnf4a as a critical gene. The authors demonstrated that OFs bind to hepatocyte lectin receptor ASGPR1/2 and increase hnf4 expression by activating JAK2/STAT3 signaling pathway. Furthermore, the authors suggest that OFs protect against infections and also increase the activity of hepatocytes. 

The authors presented a well-executed study and a nicely written manuscript. However, I have some concerns which should be addressed: 

  1. Line number 99: Authors should provide the accession numbers of data deposited in public repositories (e.g., GEO). The authors should also provide more details about how these datasets were analyzed. For example, data processing, normalization before identifying differentially expressed genes. 
  2. Line number 188 – 189: The authors identified 124 genes as differentially expressed by comparing two batches. Why differentially expressed genes from two batches are not overlapping well? From Fig. 1A seems that authors are using one set of OF treated data and comparing it with two types of control and obtaining very low overlap. The same comment goes for downregulated genes where two batches shared only 114 genes. 
  3. It is not clear how many genes authors picked from transcriptomics data to test using qPCR. 
  4. Figure 3: Description of Panel C is not included in the Figure legend. Also, Panel I not labeled in Figure. 
  5. Figure legend 3 and 4: What do the authors mean by “more fish”?
  6. “From the network analysis, we identified that zebrafish hnf4a is the driver gene that is picked up as functioning as a transcription factor by NetworkAnalyst.” How did the authors perform network analysis and identify hnf4a is not clear and should be described in the manuscript. 

Author Response

Reviewer #3 Comments: [In the presented study, Cheng et al. investigated the effect of oligo-fucoidan (OF) (extracted from brown seaweed) in hepatocytes. The authors used zebrafish treated with oligo-fucoidan and perform transcriptomics study. The differentially expressed genes identified from transcriptomics data were further confirmed with the qRT PCR. Next, using network analysis, the authors had picked hnf4a as a critical gene. The authors demonstrated that OFs bind to hepatocyte lectin receptor ASGPR1/2 and increase hnf4 expression by activating JAK2/STAT3 signaling pathway. Furthermore, the authors suggest that OFs protect against infections and also increase the activity of hepatocytes.

The authors presented a well-executed study and a nicely written manuscript. However, I have some concerns which should be addressed:

Line number 99: Authors should provide the accession numbers of data deposited in public repositories (e.g., GEO). The authors should also provide more details about how these datasets were analyzed. For example, data processing, normalization before identifying differentially expressed genes.

RESPONSE: [Thank you very much for your comments. We have provided the accession number for GEO submission, and the details for the analysis of the datasets.

The raw data of the microarray have been submitted to the NCBI Gene Expression Omnibus (GEO) (http://www.ncbi.nlm.nih.gov/geo/) under accession code GSE148324. Transcriptome Analysis Console (TAC)” software (version 4.0.2) (Affymatrix, Santa Clara, CA, USA), was used for the data processing including normalization, probe summarization, and data quality control. Guanine Cytosine Count Normalization (GCCN) and Signal Space Transformation (SST) algorithms were used for normalization to adjust CEL file intensities, allowing inter-platform comparisons. After we used TAC for data interpretation, differentially expressed genes of OF were identified for both the treated and control samples. The expression analysis settings were as follows: Gene-level fold change <−2 or >2, gene-level p-value <0.05, ANOVA method: Ebayes. ]

Line number 188 – 189: The authors identified 124 genes as differentially expressed by comparing two batches. Why differentially expressed genes from two batches are not overlapping well? From Fig. 1A seems that authors are using one set of OF treated data and comparing it with two types of control and obtaining very low overlap. The same comment goes for downregulated genes where two batches shared only 114 genes.

RESPONSE: [Thank you for your comments. We used wild-type zebrafish at 4 month of age oral feeding oligo-fucoidan for one month and then sacrificed the fish for getting liver specimens for extraction RNA and microarray analysis, we performed two batches at different time and comparing the differential expressed genes. We found only 124 upregulated genes, and 114 downregulated genes were overlapping from two batches, the reasons could be fucoidan has many bioactivities, from anti-inflammation, anti-oxidant, immunoregulation, anti-viral and anti-coagulant. Especially to the wild-type fish cultured in different time period, might experience different stress, therefore, fucoidan might differentially regulated genes with individual specificity. So, we compared two batches trying to find out the most common differentially regulated genes. I have added this part in the Discussion. ]

It is not clear how many genes authors picked from transcriptomics data to test using qPCR.

RESPONSE: [We verified three upregulated genes and five downregulated genes from transcriptomics using qPCR. We have edited the section 3.1 as following:

We verified mxf, hamp, and irak3 which were upregulated by OF (Figures 3), and five genes (fasn, scd, loxl2a, foxo3b and soat1) which were downregulated by OF (Figures 4). The gene expression in Fig. 3 echoes to the findings of Fig. 1B, and the gene expression in Fig. 4 echoes to the findings of Fig. 2B. ]

Figure 3: Description of Panel C is not included in the Figure legend. Also, Panel I not labeled in Figure.

RESPONSE: [Thank you for the correction. We have added “C” in the Figure 3 legend, and added “I” on Figure 4. ]

Figure legend 3 and 4: What do the authors mean by “more fish”?

RESPONSE: [Thank you for the comment. We have correct to qPCR for the “all” fish treated with OF. ]

“From the network analysis, we identified that zebrafish hnf4a is the driver gene that is picked up as functioning as a transcription factor by NetworkAnalyst.” How did the authors perform network analysis and identify hnf4a is not clear and should be described in the manuscript.

RESPONSE: [Thank you for the comment. We have putting all the differentially expressed genes in the NetworkAnalyst software, and found hnf4a is the driver genes for most of the differentially expressed genes. We have revised the Section 3.2 to make it clearer.

We have putting all the differentially expressed genes in the NetworkAnalyst software, and found hnf4a is the driver genes for most of the differentially expressed genes. Interestingly, hnf4a is a transcription factor in liver, and in a previous study demonstrated that the expression of Hnf4a is decreased in hepatic cirrhosis, but the overexpression of Hnf4a resets the transcription factor network in hepatocytes and prevents hepatic failure [28]. ]

Round 2

Reviewer 1 Report

The corrections have been done properly. I have no other comments.

Author Response

Thank you so much for your support.